# Subpath Queries on Compressed Graphs: A Survey

## Nicola Prezza 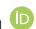

Department of Environmental Sciences, Informatics and Statistics, Ca' Foscari University, Dorsoduro, 3246, 30123 Venice, Italy; nicola.prezza@unive.it

**Abstract:** Text indexing is a classical algorithmic problem that has been studied for over four decades: given a text $T$, pre-process it off-line so that, later, we can quickly count and locate the occurrences of any string (the query pattern) in $T$ in time proportional to the query's length. The earliest optimal-time solution to the problem, the suffix tree, dates back to 1973 and requires up to two orders of magnitude more space than the plain text just to be stored. In the year 2000, two breakthrough works showed that efficient queries can be achieved without this space overhead: a fast index be stored in a space proportional to the text's entropy. These contributions had an enormous impact in bioinformatics: today, virtually any DNA aligner employs compressed indexes. Recent trends considered more powerful compression schemes (dictionary compressors) and generalizations of the problem to labeled graphs: after all, texts can be viewed as labeled directed paths. In turn, since finite state automata can be considered as a particular case of labeled graphs, these findings created a bridge between the fields of compressed indexing and regular language theory, ultimately allowing to index regular languages and promising to shed new light on problems, such as regular expression matching. This survey is a gentle introduction to the main landmarks of the fascinating journey that took us from suffix trees to today's compressed indexes for labeled graphs and regular languages.

**Keywords:** indexing; compressed data structures; labeled graphs

## 1. Introduction

Consider the classic algorithmic problem of finding the occurrences of a particular string $\Pi$ (a *pattern*) in a text $\mathcal{T}$. Classic algorithms, such as Karp-Rabin's [1], Boyer-Moore-Galil's [2], Apostolico-Giancarlo's [3], and Knuth-Morris-Pratt's [4], are optimal (the first only in the expected case) when both the text and the pattern are part of the query: those algorithms scan the text and find all occurrences of the pattern in linear time. What if the text is known *beforehand* and only the patterns to be found are part of the query? In this case, it is conceivable that *preprocessing* $\mathcal{T}$ off-line into a *fast* and *small* data structure (an *index*) might be convenient over the above on-line solutions. As it turns out, this is exactly the case. The *full-text indexing* problem (where *full* refers to the fact that we index the full set of $\mathcal{T}$'s substrings) has been studied for over forty years and has reached a very mature and exciting point: modern algorithmic techniques allow us to build text indexes taking a space close to that of the compressed text and able to count/locate occurrences of a pattern inside it in time proportional to the pattern's length. Note that we have emphasized two fundamental features of these data structures: query *time* and index *space*. As shown by decades of research on the topic, these two dimensions are, in fact, strictly correlated: the structure exploited to compress text is often the same that can be used also to support fast search queries on it. Recent research has taken a step forward, motivated by the increasingly complex structure of modern massive datasets: texts can be viewed as directed labeled path graphs and compressed indexing techniques can be generalized to more complex graph topologies. While compressed text indexing has already been covered in the literature in excellent books [5,6] and surveys [7–9], the generalizations of these advanced techniques to labeled graphs, dating back two decades, lack a single point of reference despite having reached a mature state-of-the-art. The goal of this survey is to

introduce the (non-expert) reader to the fascinating field that studies compressed indexes for labeled graphs.

We start, in Section 2, with a quick overview of classical compressed text indexing techniques: compressed suffix arrays and indexes for repetitive collections. This section serves as a self-contained warm-up to fully understand the concepts contained in the next sections. Section 3 starts with an overview of the problem of compressing graphs, with a discussion on known lower bounds to the graph indexing problem, and with preliminary solutions based on hypertext indexing. We then introduce prefix sorting and discuss its extensions to increasingly complex graph topologies. Section 3.5 is devoted to the problem of indexing trees, the first natural generalization of classical labeled paths (strings). Most of the discussion in this section is spent on the eXtended Burrows-Wheeler Transform (XBWT), a tree transformation reflecting the co-lexicographic order of the root-to-node paths on the tree. We discuss how this transformation naturally supports subpath queries and compression by generalizing the ideas of Section 2. Section 3.6 adds further diversity to the set of indexable graph topologies by discussing the cases of sets of disjoints cycles and de Bruijn graphs. All these cases are finally generalized in Section 3.7 with the introduction of *Wheeler graphs*, a notion capturing the idea of totally-sortable labeled graph. This section discusses the problem of compressing and indexing Wheeler graphs, as well as recognizing and sorting them. We also spend a paragraph on the fascinating bridge that this technique builds between compressed indexing and regular language theory by briefly discussing the elegant properties of *Wheeler languages*: regular languages recognized by finite automata in which state transition is a Wheeler graph. In fact, we argue that a useful variant of graph indexing is *regular language indexing*: in many applications (for example, computational pan-genomics [10]) one is interested in indexing the set of strings read on the paths of a labeled graphs, rather than indexing a fixed graph topology. As a matter of fact, we will see that the complexity of those two problems is quite different. Finally, Section 3.8 further generalizes prefix-sorting to any labeled graph, allowing us to index any regular language. The key idea of this generalization is to abandon total orders (of prefix-sorted states) in favor of partial ones. We conclude our survey, in Section 4, with a list of open challenges in the field.

*Terminology*

A string $S$ of length $n$ over alphabet $\Sigma$ is a sequence of $n$ elements from $\Sigma$. We use the notation $S[i]$ to indicate the $i$-th element of $S$, for $1 \leq i \leq n$. Let $a \in \Sigma$ and $S \in \Sigma^*$. We write $Sa$ to indicate the concatenation of $S$ and $a$. A string can be interpreted as an edge-labeled graph (a path) with $n + 1$ nodes connected by $n$ labeled edges. In general, a *labeled graph* $G = (V, E, \Sigma, \lambda)$ is a directed graph with set of nodes $V$, set of edges $E \subseteq V \times V$, alphabet $\Sigma$, and labeling function $\lambda : E \rightarrow \Sigma$. Quantities $n = |V|$ and $e = |E|$ will indicate the number of nodes and edges, respectively. We assume to work with an *effective* alphabet $\Sigma$ of size $\sigma$, that is, every $c \in \Sigma$ is the label of some edge. In particular, $\sigma \leq e$. We moreover assume the alphabet to be totally ordered by an order we denote with $\leq$, and write $a < b$ when $a \leq b$ and $a \neq b$. In this survey, we consider two extensions of $\leq$ to strings (and, later, to labeled graphs). The *lexicographic order* of two strings $aS$ and $a'S'$ is defined as $a'S' < aS$ if and only if either (i) $a' < a$ or (ii) $a = a'$ and $S' < S$ hold. The empty string $\epsilon$ is always smaller than any non-empty string. Symmetrically, the *co-lexicographic order* of two strings $Sa$ and $S'a'$ is defined as $S'a' < Sa$ if and only if either (i) $a' < a$ or (ii) $a = a'$ and $S' < S$ hold.

This paper deals with the indexed pattern matching problem: preprocess a text so that, later all text occurrences of any query pattern $\Pi \in \Sigma^m$ of length $m$ can be efficiently counted and located. These queries can be generalized to labeled graphs; we postpone the exact definition of this generalization to Section 3.

## 2. The Labeled Path Case: Indexing Compressed Text

Consider the text reported in Figure 1. For reasons that will be made clear later, we

append a special symbol \$ at the end of the text, and assume that \$ is lexicographically smaller than all other characters. Note that we assign a different color to each distinct letter.

$$\mathcal{T} \quad = \quad A \quad T \quad A \quad T \quad A \quad G \quad A \quad T \quad \$$$
$$\phantom{\mathcal{T} \quad = \quad} 1 \quad 2 \quad 3 \quad 4 \quad 5 \quad 6 \quad 7 \quad 8 \quad 9$$

**Figure 1.** Running example used in this section.

The question is: how can we design a data structure that permits to efficiently find the exact occurrences of (short) strings inside $\mathcal{T}$? In particular, one may wish to *count* the occurrences of a pattern (for example, $count(\mathcal{T}, "AT") = 3$) or to *locate* them (for example, $locate(\mathcal{T}, "AT") = 1, 3, 7$). As we will see, there are two conceptually different ways to solve this problem. The first, sketched in Section 2.1, is to realize that every occurrence of $\Pi = "AT"$ in $\mathcal{T}$ is a prefix of a text suffix (for example: occurrence at position 7 of "AT" is a prefix of the text suffix "AT"). The second, sketched in Section 2.2, is to partition the text into non-overlapping *phrases* appearing elsewhere in the text and divide the problem into the two sub-problems of (i) finding occurrences that overlap two adjacent phrases and (ii) finding occurrences entirely contained in a phrase. Albeit conceptually different, both approaches ultimately resort to *suffix sorting*: sorting lexicographically a subset of the text's suffixes. The curious reader can refer to the excellent reviews of Mäkinen and Navarro [7] and Navarro [8,9] for a much deeper coverage of the state-of-the-art relative to both approaches.

### 2.1. The Entropy Model: Compressed Suffix Arrays

The sentence *every occurrence of $\Pi$ is the prefix of a suffix of $\mathcal{T}$* leads very quickly to a simple and time-efficient solution to the full-text indexing problem. Note that a suffix can be identified by a text position: for example, suffix "GAT\$" corresponds to position 6. Let us sort text suffixes in *lexicographic order*. We call the resulting integer array the *suffix array* (SA) of $\mathcal{T}$; see Figure 2. This time, we color *text positions i* according to the color of letter $\mathcal{T}[i]$. Note that colors (letters) get clustered, since suffixes are sorted lexicographically.

$$
\begin{array}{ccccccccccc}
SA & = & 9 & 5 & 7 & 3 & 1 & 6 & 8 & 4 & 2 \\
   &   & \$ & A & A & A & A & G & T & T & T \\
   &   &    & G & T & T & T & \$ & A & A & A \\
   &   &    & A & \$ & A & A & T &   & G & T \\
   &   &    & T &    & G & T & \$ &   & A & A \\
   &   &    & \$ &   & A & A &    &   & T & G \\
   &   &    &    &   & T & G &    &   & \$ & A \\
   &   &    &    &   & \$ & A &    &   &    & T \\
   &   &    &    &   &    & T &    &   &    & \$ \\
   &   &    &    &   &    & \$ &    &   &    &    \\
\end{array}
$$

**Figure 2.** Suffix Array (SA) of the text $\mathcal{T}$ of Figure 1. Note: we store only array $SA$ and the text $\mathcal{T}$, not the actual text suffixes.

The reason for appending \$ at the end of the text is that, in this way, no suffix prefixes another suffix. Suffix arrays were independently discovered by Udi Manber and Gene Myers in 1990 [11] and (under the name of PAT array) by Gonnet, Baeza-Yates, and Snider in 1992 [12,13]. An earlier solution, the *suffix tree* [14], dates back to 1973 and is more time-efficient, albeit much less space efficient (by a large constant factor). The idea behind the suffix tree is to build the trie of all text's suffixes, replacing unary paths with pairs of pointers to the text.

The suffix array $SA$, when used together with the text $\mathcal{T}$, is a full-text index: in order to count/locate occurrences of a pattern $\Pi$, it is sufficient to binary search $SA$, extracting characters from $\mathcal{T}$ to compare $\Pi$ with the corresponding suffixes during search. This solution permits to count the *occ* occurrences of a pattern $\Pi$ of length $m$ in a text $\mathcal{T}$ of length $n$ in $O(m \log n)$ time, as well as to report them in additional optimal $O(occ)$ time.

Letting the alphabet size being denoted by $\sigma$, our index requires $n \log \sigma + n \log n$ bits to be stored (the first component for the text and the second for the suffix array).

Note that the suffix array cannot be considered a *small* data structure. On a constant-sized alphabet, the term $n \log n$ is asymptotically larger than the text. The need for processing larger and larger texts made this issue relevant by the end of the millennium. As an instructive example, consider indexing the Human genome ($\approx$3 billion DNA bases). DNA sequences can be modeled as strings over an alphabet of size four ($\{A, C, G, T\}$); therefore, the Human genome takes less than 750 MiB of space to be stored using two bits per letter. The suffix array of the Human genome requires, on the other hand, about 11 GiB.

Although the study of *compressed text indexes* had begun before the year 2000 [15], the turn of the millennium represented a crucial turning point for the field: two independent works by Ferragina and Manzini [16] and Grossi and Vitter [17] showed how to *compress* the suffix array.

Consider the integer array $\psi$ of length $n$ defined as follows. Given a suffix array position $i$ containing value (text position) $j = SA[i]$, the cell $\psi[i]$ contains the suffix array position $i'$ containing text position $(j \mod n) + 1$ (we treat the string as circular). More formally: $\psi[i] = SA^{-1}[(SA[i] \mod n) + 1]$. See Figure 3.

$$
\begin{array}{ccccccccccc}
SA & = & 9 & 5 & 7 & 3 & 1 & 6 & 8 & 4 & 2 \\
\psi & = & 5 & 6 & 7 & 8 & 9 & 3 & 1 & 2 & 4 \\
 & & 1 & 2 & 3 & 4 & 5 & 6 & 7 & 8 & 9
\end{array}
$$

**Figure 3.** Array $\psi$.

Note the following interesting property: equally-colored values in $\psi$ are increasing. More precisely: $\psi$-values corresponding to suffixes beginning with the same letter form increasing subsequences. To understand why this happens, observe that $\psi[i]$ takes us from a suffix $\mathcal{T}[SA[i], \ldots, n]$ to suffix $\mathcal{T}[SA[i] + 1, \ldots, n]$ (let us exclude the case $SA[i] = n$ in order to simplify our formulas). Now, assume that $i < j$ and $\mathcal{T}[SA[i], \ldots, n]$ and $\mathcal{T}[SA[j], \ldots, n]$ begin with the same letter (that is: $i$ and $j$ have the same color in Figure 3). Since $i < j$, we have that $\mathcal{T}[SA[i], \ldots, n] < \mathcal{T}[SA[j], \ldots, n]$ (lexicographically). But then, since the two suffixes begin with the same letter, we also have that $\mathcal{T}[SA[i] + 1, \ldots, n] < \mathcal{T}[SA[j] + 1, \ldots, n]$, i.e., $\psi[i] < \psi[j]$.

Let us now store just the differences between consecutive values in each increasing subsequence of $\psi$. We denote this new array of differences as $\Delta(\psi)$. See Figure 4 for an example.

$$
\begin{array}{ccccccccccc}
SA & = & 9 & 5 & 7 & 3 & 1 & 6 & 8 & 4 & 2 \\
\Delta(\psi) & = & 5 & 6 & 1 & 1 & 1 & 3 & 1 & 1 & 2 \\
 & & 1 & 2 & 3 & 4 & 5 & 6 & 7 & 8 & 9
\end{array}
$$

**Figure 4.** The differential array $\Delta(\psi)$.

Two remarkable properties of $\Delta(\psi)$ are that, using an opportune encoding for its elements, this sequence:

1.  supports accessing any $\psi[i]$ in constant time [17], and
2.  can be stored in $nH_0 + O(n)$ bits of space, where $H_0 = \sum_{c \in \Sigma} (n_c/n) \log(n/n_c)$ is the zero-order empirical entropy of $\mathcal{T}$ and $n_c$ denotes the number of occurrences of $c \in \Sigma$ in $\mathcal{T}$.

The fact that this strategy achieves compression is actually not hard to prove. Consider any integer encoding (for example, Elias' delta or gamma) capable to represent any integer $x$ in $O(1 + \log x)$ bits. By the concavity of the logarithm function, the inequality $\sum_{i=1}^{m} \log(x_i) \leq m \cdot \log\left(\frac{\sum_{i=1}^{m} x_i}{m}\right)$ holds for any integer sequence $x_1, \ldots, x_m$. Now, note that the sub-sequence $x_1, \ldots, x_{n_c}$ of $\Delta(\psi)$ corresponding to letter $c \in \Sigma$ has two properties: it has exactly $n_c$ terms and $\sum_{i=1}^{n_c} x_i \leq n$. It follows that the encoded sub-sequence takes (asymptotically) $\sum_{i=1}^{n_c} (1 + \log(x_i)) \leq n_c \cdot \log(n/n_c) + n_c$ bits. By summing this quantity

over all the alphabet's characters, we obtain precisely $O(n(H_0 + 1))$ bits. Other encodings (in particular, Elias-Fano dictionaries [18,19]) can achieve the claimed $nH_0 + O(n)$ bits while supporting constant-time random access.

The final step is to recognize that $\psi$ moves us forward (by one position at a time) in the text. This allows us to extract suffixes *without using the text*. To achieve this, it is sufficient to store in one array $F = \$AAAAGTTT$ (using our running example) the first character of each suffix. This can be done in $O(n)$ bits of space (using a bitvector) since those characters are sorted and we assume the alphabet to be effective. The *i*-th text suffix (in lexicographic order) is then $F[i], F[\psi[i]], F[\psi^{(2)}[i]], F[\psi^{(3)}[i]], \ldots$, where $\psi^{(\ell)}$ indicates function $\psi$ applied $\ell$ times to its argument. See Figure 5. Extracting suffixes makes it possible to implement the binary search algorithm discussed in the previous section: the *Compressed Suffix Array* (CSA) takes compressed space and enables us finding all pattern's occurrences in a time proportional to the pattern length. By adding a small sampling of the suffix array, one can use the same solution to compute any value $SA[i]$ in polylogarithmic time without asymptotically affecting the space usage.

|           | 1   | 2   | 3   | 4   | 5   | 6   | 7   | 8   | 9   |
|-----------|-----|-----|-----|-----|-----|-----|-----|-----|-----|
| $\psi$ =  | 5   | 6   | 7   | 8   | 9   | 3   | 1   | 2   | 4   |
| $F$ =     | $\$$ | $A$ | $A$ | $A$ | $A$ | $G$ | $T$ | $T$ | $T$ |
|           |     | $G$ | $T$ | $T$ | $T$ | $A$ | $\$$ | $A$ | $A$ |
|           |     | $A$ | $\$$ | $A$ | $A$ | $T$ |     | $G$ | $T$ |
|           |     | $T$ |     | $G$ | $T$ | $\$$ |     | $A$ | $A$ |
|           |     | $\$$ |     | $A$ | $A$ |     |     | $T$ | $G$ |
|           |     |     |     | $T$ | $G$ |     |     | $\$$ | $A$ |
|           |     |     |     | $\$$ | $A$ |     |     |     | $T$ |
|           |     |     |     |     | $T$ |     |     |     | $\$$ |
|           |     |     |     |     | $\$$ |     |     |     |     |

**Figure 5.** Compressed Suffix Array (CSA): we store the delta-encoded $\psi$ and the first letter (underlined) of each suffix (array F).

Another (symmetric) philosophy is to exploit the inverse of function $\psi$. These indexes are based on the *Burrows-Wheeler transform* (BWT) [20] and achieve high-order compression [16]. We do not discuss BWT-based indexes here since their generalizations to labeled graphs will be covered in Section 3. For more details on entropy-compressed text indexes, we redirect the curious reader to the excellent survey of Mäkinen and Navarro [7].

### 2.2. The Repetitive Model

Since their introduction in the year 2000, entropy-compressed text indexes have had a dramatic impact in domains, such as bioinformatics; the most widely used DNA aligners, Bowtie [21] and BWA [22], are based on compressed indexes and can align thousands of short DNA fragments per second on large genomes while using only compressed space in RAM during execution. As seen in the previous section, these indexes operate within a space bounded by the text's empirical entropy [16,17]. Entropy, however, is insensitive to long repetitions: the entropy-compressed version of $\mathcal{T} \cdot \mathcal{T}$ (that is, text $\mathcal{T}$ concatenated with itself) takes at least twice the space of the entropy-compressed $\mathcal{T}$ [23]. There is a simple reason for this fact: by its very definition (see Section 2.1), the quantity $H_0$ depends only on the characters' relative frequencies in the text. Since characters in $\mathcal{T}$ and $\mathcal{T} \cdot \mathcal{T}$ have the same relative frequencies, it follows that their entropies are the same. The claim follows, being the length of $\mathcal{T} \cdot \mathcal{T}$ twice the length of $\mathcal{T}$.

While the above reasoning might seem artificial, the same problem arises on texts composed of large repetitions. Today, most large data sources, such as DNA sequencers and the web, follow this new model of *highly repetitive data*. As a consequence, entropy-compressed text indexes are no longer able to keep pace with the exponentially-increasing rate at which this data is produced, as very often the mere index size exceeds the RAM limit. For this reason, in recent years more powerful compressed indexes have emerged; these

are based on the Lempel-Ziv factorization [23], the run-length Burrows-Wheeler Transform (BWT) [24–26], context-free grammars [27], string attractors [28,29] (combinatorial objects generalizing those compressors), and more abstract measures of repetitiveness [30]. The core idea behind these indexes (with the exception of the run-length BWT; see Section 3) is to partition the text into phrases that occur also elsewhere and to use *geometric* data structures to locate pattern occurrences.

To get a more detailed intuition of how these indexes work, consider the Lempel-Ziv'78 (LZ78) compression scheme. The LZ78 factorization breaks the text into phrases with the property that each phrase extends by one character a previous phrase; see the example in Figure 6 (top). The mechanism we are going to describe works for any factorization-based compressor (also called *dictionary compressors*). A factorization-based index keeps a two-dimensional geometric data structure storing a labeled point $(x, y, \ell)$ for each phrase $y$, where $x$ is the reversed text prefix preceding the phrase and $\ell$ is the first text position of the phrase. Efficient techniques not discussed here (in particular, mapping to *rank space*) exist to opportunely reduce $x$ and $y$ to small integers preserving the lexicographic order of the corresponding strings (see, for example, Kreft and Navarro [23]). For example, such a point in Figure 6 is $(GCA, CG, 4)$, corresponding to the phrase break between positions 3 and 4. To understand how *locate* queries are answered, consider the pattern $\Pi = CAC$. All occurrences of $\Pi$ crossing a phrase boundary can be split into a prefix and a suffix, aligned on the phrase boundary. Let us consider the split $CA|C$ (all possible splits have to be considered for the following algorithm to be complete). If $\Pi$ occurs in $\mathcal{T}$ with this split, then $C$ will be a prefix of some phrase $y$, and $CA$ will be a suffix of the text prefix preceding $y$. We can find all phrases $y$ with this property by issuing a four-sided range query on our grid as shown in Figure 6 (bottom left). This procedure works since $C$ and $AC$ (that is, $CA$ reversed) define ranges on the horizontal and vertical axes: these ranges contain all phrases prefixed by $C$ and all reversed prefixes prefixed by $AC$ (equivalently, all prefixes suffixed by $CA$), respectively. In Figure 6 (bottom left), the queried rectangle contains text positions 13 and 11. By subtracting from those numbers the length of the pattern's prefix (in this example, $2 = |CA|$), we discover that $\Pi = CAC$ occurs at positions 11 and 9 crossing a phrase with split $CA|C$.

With a similar idea (that is, resorting again to geometric data structures), one can recursively track occurrences completely contained inside a phrase; see the caption of Figure 6 (bottom right). Let $b$ be the number of phrases in the parse. Note that our geometric structures store overall $O(b)$ points. On repetitive texts, the smallest possible number $b$ of phrases of such a parse can be *constant*. In practice, on repetitive texts $b$ (or any of its popular approximations [31]) is orders of magnitude smaller than the entropy-compressed text [23,24]. The fastest existing factorization-based indexes to date can locate all *occ* occurrences of a pattern $\Pi \in \Sigma^m$ in $O(b \text{ polylog } n)$ bits of space and *optimal* $O(m + occ)$ time [32]. For more details on indexes for repetitive text collections, the curious reader can refer to the excellent recent survey of Navarro [8,9].

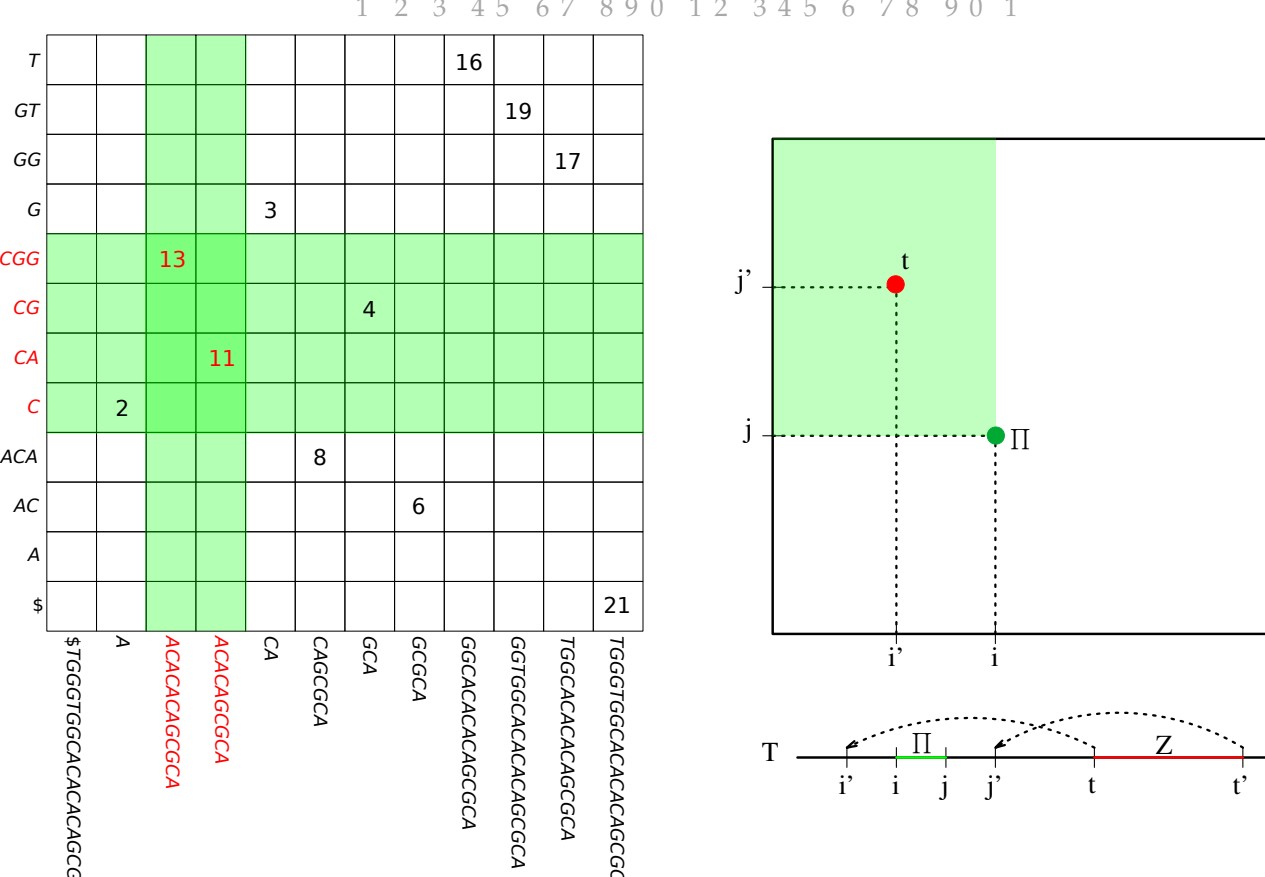

**Figure 6.** (**Top**) Parsed text according to the LZ78 factorization. Text indices are shown in gray. (**Bottom left**) four-sided geometric data structure storing one labeled point $(x, y, \ell)$ per phrase $y$, where $x$ is the reversed text prefix preceding phrase $y$ and $\ell$ is the first position of the phrase. (**Bottom right**) given a pattern occurrence $\Pi = \mathcal{T}[i, j]$ (green dot), we can locate all phrases that completely copy it. For each phrase $\mathcal{T}[t, t + \ell - 1]$ for which source is $\mathcal{T}[i', i' + \ell - 1]$, a labeled point $(i', i' + \ell - 1, t)$ is inserted in the data structure. In the example: phrase $Z = \mathcal{T}[t, t']$, red dot, copies $\mathcal{T}[i', j']$ which completely contains $\Pi$. Note that $\Pi = \mathcal{T}[i, j]$ defines the query, while each phrase generates a point that is stored permanently in the geometric structure.

## 3. Indexing Labeled Graphs and Regular Languages

Interestingly enough, most techniques seen in the previous section extend to more structured data. In particular, in this section, we will work with *labeled graphs*, and discuss extensions of these results to *regular languages* by interpreting finite automata as labeled graphs. We recall that a labeled graph is a quadruple $(V, E, \Sigma, \lambda)$ where $V$ is a set of $n = |V|$ nodes, $E \subseteq V \times V$ is a set of $e = |E|$ directed edges, $\Sigma$ is the alphabet and $\lambda : E \to \Sigma$ is a labeling function assigning a label to each edge. Let $P = (u_{i_1}, u_{i_2}), (u_{i_2}, u_{i_3}), \dots, (u_{i_k}, u_{i_{k+1}})$ be a path of length $k$. We extend function $\lambda$ to paths as $\lambda(P) = \lambda((u_{i_1}, u_{i_2})) \cdot \lambda((u_{i_2}, u_{i_3})) \cdots \lambda((u_{i_k}, u_{i_{k+1}}))$, and say that $\lambda(P)$ is the string labeling path $P$. A node $u$ is *reached by a path labeled* $\Pi \in \Sigma^*$ if there exists a path $P = (u_{i_1}, u_{i_2}), (u_{i_2}, u_{i_3}), \dots, (u_{i_k}, u)$ ending in $u$ such that $\lambda(P) = \Pi$.

The starting point is to observe that texts are nothing but labeled *path graphs*. As it turns out, there is nothing special about paths that we cannot generalize to more complex topologies. We adopt the following natural generalizations of count and locate queries:

- *Count*: given a pattern (a string) $\Pi \in \Sigma^m$, return the number of nodes reached by a path labeled with $\Pi$.

- *Locate*: given a pattern Π, return a representation of all nodes reached by a path labeled with Π.

We will refer to the set of these two queries with the term *subpath queries*. The node representation returned by locate queries could be an arbitrary numbering. This solution, however, has the disadvantage of requiring at least $n \log n$ bits of space just for the labels, $n$ being the number of nodes. In the following subsections, we will discuss more space-efficient solutions based on the idea of returning a more regular labeling (e.g., the DFS order of the nodes).

### 3.1. Graph Compression

We start with a discussion of existing techniques for compressing labeled graphs. Lossless graph compression is a vast topic that has been treated more in detail in other surveys [33,34]. Since here we deal with subpath queries on compressed graphs, we only discuss compression techniques that have been shown to support these queries on special cases of graphs (mainly paths and trees).

Note that, differently from the string domain, the goal is now to compress two components: the labels and the graph's topology. Labels can be compressed by extending the familiar entropy model to graphs. The idea here is to simply count the frequencies of each character in the multi-set of all the edges' labels. By using an encoding, such as Huffman's, the zero-order empirical entropy of the labels can be approached. One can take a step further and compress labels to their high-order entropy. As noted by Ferragina et al. [35], if the topology is a tree then we can use a different zero-order model for each context of length $k$ (that is, for each distinct labeled path of length $k$) preceding a given edge. This is the same observation used to compress texts: it is much easier to predict a particular character if we know the $k$ letters (or the path of length $k$) that precede it. The entropy model is already very effective in compressing the labels component. The most prominent example of entropy-compressed tree index is the eXtended Burrows Wheeler transform [35], covered more in detail in Section 3.5. As far as the topology is concerned, things get more complicated. The topology of a tree with $n$ nodes can be represented in $2n$ bits via its balanced-parentheses representation. Ferres et al. [36] used this representation and the idea that planar graphs are fully specified by a spanning tree of the graph and one of its dual to represent planar graphs in just $4n$ bits per node. Recently, Chakraborty et al. proposed succinct representations for finite automata [37]. General graph topologies are compressed well, but without worst-case guarantees, by techniques, such as $K^2$ trees [38]. Further compression can be achieved by extending the notion of entropy to the graph's topology, as shown by Jansson et al. [39], Hucke et al. [40], and Ganczorz [41] for the particular case of trees. As it happens with text entropy, their measures work well under the assumption that the tree topology is not extremely repetitive. See Hucke et al. [42] for a systematic comparison of tree entropy measures. In the repetitive case, more advanced notions of compression need to be considered. Straight-line grammars [43,44], that is, context-free grammars generating a single graph, are the gold-standard for capturing such repetitions. These representations can be augmented to support constant-time traversal of the compressed graph [45] but do not support indexing queries on topologies being more complex than paths. Other techniques (defined only for trees) include the Lempel-Ziv factorization of trees [46] and top trees [47]. Despite being well-suited for compressing repetitive tree topologies, these representations also are not (yet) able to support efficient indexing queries on topologies more complex than paths so we do not discuss them further. The recent *tunneling* [48] and *run-length XBW Transform* [49] compression schemes are the first compressed representations for repetitive topologies supporting count queries (the latter technique supports also locate queries, but it requires additional linear space). These techniques can be applied to Wheeler graphs [50], as well, and are covered more in detail in Sections 3.5 and 3.7.

### 3.2. Conditional Lower Bounds

Before diving into compressed indexes for labeled graphs, we spend a paragraph on the complexity of matching strings on labeled graphs. The main results in this direction are due to Backurs and Indyk [51] and to Equi et al. [52–54], and we considered the on-line version of the problem: both pre-processing and query time are counted towards the total running time. The former work [51] proved that, unless the Strong Exponential Time Hypothesis (SETH) [55] is false, *in the worst case* no algorithm can match a regular expression of size $e$ against a string of length $m$ in time $O((m \cdot e)^{1-\delta})$, for any constant $\delta > 0$. Since regular expressions can be converted into NFAs of the same asymptotic size, this result implies a quadratic conditional lower-bound to the graph pattern matching problem. Equi et al. [54] improved this result to include graphs of maximum node degree equal to two and deterministic directed acyclic graphs. Another recent paper from Potechin and Shallit [56] established a similar hardness result for the problem of determining whether a NFA accepts a given word: they showed that, provided that the NFA is sparse, a sub-quadratic algorithm for the problem would falsify SETH. All these results are based on reductions from the orthogonal vectors problem: find two orthogonal vectors in two given sets of binary $d$-dimensional vectors (one can easily see the similarity between this problem and string matching). The orthogonal vectors hypothesis (OV) [57] states that the orthogonal vectors problem cannot be solved in strongly subquadratic time. A further improvement has recently been made by Gibney [58], who proved that even shaving logarithmic factors from the running time $O(m \cdot e)$ would yield surprising new results in complexity theory, as well as falsifying an hypothesis recently made by Abboud and Bringmann in Reference [59] on the fine-grained complexity of SAT solvers. As noted above, in the context of our survey, these lower bounds imply that the sum between the construction and query times for a graph index cannot be sub-quadratic unless important conjectures in complexity theory fail. These lower bounds, however, do not rule out the possibility that a graph index could support efficient (say, subquadratic) queries at the cost of an expensive (quadratic or more) index construction algorithm. The more recent works of Equi et al. [52,53] addressed precisely this issue: they proved that no index that can be built in polynomial time ($O(e^\alpha)$ for any constant $\alpha \geq 1$) can guarantee strongly sub-quadratic query times, that is, $O(e^\delta m^\beta)$ query time for any constants $\delta < 1$ or $\beta < 1$. This essentially settles the complexity of the problem: assuming a reasonably fast (polynomial time) index construction algorithm, subpath-query times need to be at least quadratic ($O(m \cdot e)$) in the worst case. Since these bounds are matched by existing on-line algorithms [60], one may wonder what is the point of studying graph indexes at all. As we will see, the answer lies in parameterized complexity: even though pattern matching on graphs/regular expressions is hard *in the worst case*, it is indeed possible to solve the problem efficiently in particular cases (for example, trees [35] and particular regular expressions [51]). Ultimately, in Section 3.8 we will introduce a complete hierarchy of labeled graphs capturing the hardness of the problem.

### 3.3. Hypertext Indexing

Prior to the predominant prefix-sorting approach that we are going to discuss in detail in the next subsections, the problem of solving indexed path queries on labeled graphs has been tackled in the literature by resorting to geometric data structures [61,62]. These solutions work in the *hypertext model*: the objects being indexed are *node-labeled* graphs $G = (V, E, \Sigma, \lambda)$, where function $\lambda : V \to \Sigma^*$ associates a string to each node (note the difference with our edge-labeled model, where each *edge* is labeled with a *single* character). Let $P = (u_{i_1}, u_{i_2}), (u_{i_2}, u_{i_3}), \ldots, (u_{i_{k-1}}, u_{i_k})$ be a path in the graph spanning $k$ (not necessarily distinct) nodes. With $\lambda(P)$ we denote the string $\lambda(P) = \lambda(u_{i_1}) \cdot \lambda(u_{i_2}) \cdots \lambda(u_{i_k})$. In the hypertext indexing problem, the goal is to build an index over $G$ able to quickly support locate queries on the paths of $G$: given a pattern $\Pi$, determine the positions in the graph (node and starting position in the node) where an occurrence of $\Pi$ starts. This labeled graph model is well suited for applications where the strings labeling each node

are very long (for example, a transcriptome), in which case the label component (rather than the graph's topology) dominates the data structure's space. Both solutions discussed in Reference [61,62] resort to geometric data structures. First, a classic text index (for example, a compressed suffix array) is built over the concatenation $\lambda(u_1) \cdot \# \cdots \# \cdot \lambda(u_n)$ of the strings labeling all the graph's nodes $u_1, \ldots, u_n$. The labels are separated by a symbol # not appearing elsewhere in order to prevent false matches. Pattern occurrences completely contained inside some $\lambda(u_i)$ are found using this compressed index. Then, a geometric structure is used to "glue" nodes connected by an edge: for each edge $(u, v)$, a two-dimensional point $(\overleftarrow{\lambda(u)}, \lambda(v))$ is added to a grid as the one shown in Figure 6 (bottom left), where $\overleftarrow{w}$ denotes the reverse of string $w$. Again, rank space techniques are used to reduce the points $(\overleftarrow{\lambda(u)}, \lambda(v))$ to integer pairs. Then, pattern occurrences spanning *one* edge are found by issuing a four-sided geometric query for each possible pattern split as seen in Section 2.2. The main issue with these solutions is that they cannot *efficiently* locate pattern occurrences spanning two or more edges. A tweak consists in using a seed-and-extend strategy: the pattern $\Pi$ is split in short fragments of some length $t$, each of which is searched separately in the index. If $\lambda(u_i) \geq t$ for each node $u_i$, then this solution allows locating all pattern occurrences (including those spanning two or more edges). This solution, however, requires to visit the whole graph in the worst case. On realistic DNA datasets this problem is mitigated by the fact that, for large enough $t$, the pattern's fragments of length $t$ are "good anchors" and are likely to occur only within occurrences of $\Pi$.

As seen, hypertext indexes work well under the assumption that the strings labeling the nodes are very long. However, this assumption is often not realistic: for instance, in pan-genomics applications a single-point mutation in the genome of an individual (that is, a substitution or a small insertion/deletion) may introduce very short edges deviating from the population's reference genome. Note that here we have only discussed *indexed* solutions. The on-line case (pattern matching on hypertext without pre-processing) has been thoroughly studied in several other works; see Reference [60,63,64].

*3.4. Prefix Sorting: Model and Terminology*

A very natural approach to solve efficiently the graph indexing problem on arbitrary labeled graphs is to generalize suffix sorting (Actually, for technical reasons we will use the symmetric *prefix sorting*) from strings to labeled graphs. In the following, we will make a slight simplification and work with topologies corresponding to the transition function of nondeterministic finite automata (NFAs). In particular, we will assume that:

1.  there is only one node, deemed the *source state* (or start state), without incoming edges, and
2.  any state is reachable from the source state.

We will use the terms *node* and *state* interchangeably. For reasons that will become clear later, we will moreover assume that the set of characters labeling the incoming edges of any node is a singleton. These assumptions are not too restrictive. It is not hard to see that these NFAs can, in fact, recognize any regular language [65] (in particular, any NFA can be transformed in polynomial time to meet these requirements).

We make a little twist and replace the lexicographic order of suffixes with the symmetric co-lexicographic order of prefixes. This turns out to be much more natural when dealing with NFAs, as we will sort states according to the co-lexicographic order of the strings labeling paths connecting the source with each state. As additional benefits of these choices:

1.  we will be able to search strings *forward* (that is, left-to-right). In particular, this will enable testing membership of words in the language recognized by an indexed NFA in a natural on-line (character-by-character) fashion ( traditionally, BWT-based text indexes are based on suffix sorting and support *backward* search of patterns in the text).

2. We will be able to transfer powerful language-theoretic results to the field of compressed indexing. For example, a co-lexicographic variant of the Myhill-Nerode theorem [66] will allow us to minimize the number of states of indexable DFAs.

### 3.5. Indexing Labeled Trees

Kosaraju [67] was the first to extend prefix sorting to a very simple class of labeled graphs: labeled trees. To do so, he defined the suffix tree of a (reversed) trie. To understand this idea, we are now going to describe a simplified data structure: the prefix array of a labeled tree.

#### 3.5.1. The Prefix Array of a Labeled Tree

Consider the list containing the tree's nodes sorted by the co-lexicographic order of the paths connecting them to the root. We call this array the *prefix array* (PA) of the tree. See Figure 7 for an example: node 5 comes before node 8 in the ordering because the path connecting the root to 5, "aac", is co-lexicographically smaller than the path connecting the root to 8, "bc". This node ordering is uniquely defined if the tree is a trie (that is, if each node has at most one outgoing edge per label). Otherwise, the order of nodes reached by the same root-to-node path can be chosen arbitrarily.

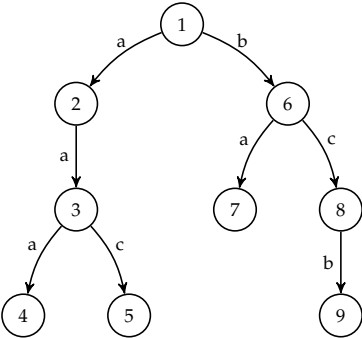

**Figure 7.** Example of labeled tree. Nodes have been enumerated in pre-order. The list of nodes sorted by the co-lexicographic order of the paths connecting them to the root (that is, the prefix array (PA) of the tree) is: 1, 2, 3, 4, 7, 6, 9, 5, 8.

A nice property of the prefix array of a tree is that, together with the tree itself, it is already an index; in fact, it is a straightforward generalization of the suffix array SA introduced in Section 2 (in this case, we call it "prefix array" since we are sorting prefixes of root-to-leaves paths). Since nodes are sorted co-lexicographically, the list of nodes reached by a path labeled with a given pattern $\Pi$ can be found by binary search. At each search step, we jump on the corresponding node on the tree and compare $\Pi$ with the labels extracted on the path connecting the node with the root. This procedure allows counting all nodes reached by $\Pi$, as well as subsequently reporting them in optimal constant time each, in $O(|\Pi| \log n)$ time, $n$ being the number of nodes.

#### 3.5.2. The XBW Transform

One disadvantage of the prefix array is that, like the SA seen in Section 2.1, it does not achieve compression. In addition to the $n \log n$ bits for the prefix array itself, the above binary-search strategy also requires us to navigate the tree (for which topology and labels must therefore be kept in memory). This is much more space than the labels, which require $n \log \sigma$ bits when stored in plain format, and the tree topology, which can be stored in just $2n$ bits (for example, in balanced-parentheses sequence representation). In 2005, Ferragina et al. [35,68] observed that this space overhead is not necessary: an efficient search machinery can be fit into a space proportional to the entropy-compressed edge labels, plus the succinct $(2n + o(n)$ bits) tree's topology. Their structure takes the name *XBW tree transform* (XBWT in the following), and is a compressed tree representation

natively supporting subpath search queries. Consider the co-lexicographic node ordering of Figure 7, denote with $child(u)$ the multi-set of outgoing labels of node $u$ (for the leaves, $child(u) = \varnothing$), and with $\lambda(u)$ the incoming label of node $u$ (for the root, $\lambda(u) = \$$, where as usual $\$$ is lexicographically smaller than all other characters). Let $u_1, \ldots, u_n$ be the prefix array of the tree. The XBWT is the pair of sequences $IN = \lambda(u_1) \ldots \lambda(u_n)$ and $OUT = child(u_1), \ldots, child(u_n)$. See Figure 8 for an example based on the tree of Figure 7. For simplicity, the tree of this example is a trie (that is, a deterministic tree). All of the following reasoning, however, applies immediately to arbitrary labeled trees.

| $i$ | 1 | 2 | 3 | 4 | 5 | 6 | 7 | 8 | 9 |
|---|---|---|---|---|---|---|---|---|---|
| PA | 1 | 2 | 3 | 4 | 7 | 6 | 9 | 5 | 8 |
| $IN$ | \$ | a | a | a | a | b | b | c | c |
| $OUT$ | a<br>b<br><br>c | a | a<br><br>c | | | a<br><br>c | | | b |

**Figure 8.** Prefix array (PA) and sequences $IN$ and $OUT$ forming the eXtended Burrows-Wheeler Transform (XBWT). $IN$ (incoming labels) is the sequence of characters labeling the incoming edge of each node, while $OUT$ (outgoing labels) is the sequence of multi-sets containing the characters labeling the outgoing edges of each node.

3.5.3. Inverting the XBWT

As it turns out, the tree can be reconstructed from just $OUT$. To prove this, we show that the XBWT representation can be used to perform a tree visit. First, we introduce the key property at the core of the XBWT: edges' labels appear in the same order in $IN$ and $OUT$, that is, the $i$-th occurrence (counting from left to right) of character $c \in \Sigma$ in $IN$ corresponds to the same edge of the $i$-th occurrence $c$ in $OUT$ (the order of characters inside each multi-set of $OUT$ is not relevant for the following reasoning to work). For example, consider the fourth 'a' in $IN$, appearing at $IN[5]$. This label corresponds to the incoming edge of node 7, that is, to edge $(6, 7)$. The fourth occurrence of 'a' in $OUT$ appears in $OUT[6]$, corresponding to the outgoing edges of node 6. By following the edge labeled 'a' from node 6, we reach exactly node 7, that is, this occurrence of 'a' labels edge $(6, 7)$. Why does this property hold? precisely because we are sorting nodes co-lexicographically. Take two nodes $u < v$ such that $\lambda(u) = \lambda(v) = a$, for example, $u = 2$ and $v = 7$ (note that $<$ indicates the co-lexicographic order, not pre-order). Since $u < v$, the $a$ labeling edge $(\pi(u), u)$ precedes the $a$ labeling edge $(\pi(v), v)$ in sequence $IN$, where $\pi(u)$ indicates the parent of $u$ in the tree. In the example, these are the two 'a's appearing at $IN[2]$ and $IN[5]$. Let $\alpha_u$ and $\alpha_v$ be the two strings labeling the two paths from the root to $u$ and $v$, respectively. In our example, $\alpha_2 = \$a$ and $\alpha_7 = \$ba$ (note that we prepend the artificial incoming label of the root). By the very definition of our co-lexicographic order, $u < v$ if and only if $\alpha_u < \alpha_v$. Note that we can write $\alpha_u = \alpha_{\pi(u)} \cdot a$ and $\alpha_v = \alpha_{\pi(v)} \cdot a$. Then, $\alpha_u < \alpha_v$ holds if and only if $\alpha_{\pi(u)} < \alpha_{\pi(v)}$, i.e., if and only if $\pi(u) < \pi(v)$. In our example, $\alpha_{\pi(2)} = \alpha_1 = \$ < \$b = \alpha_6 = \alpha_{\pi(7)}$; thus, it must hold $1 = \pi(2) < \pi(7) = 6$ (which, in fact, holds in the example). This means that the $a$ labeling edge $(\pi(u), u)$ comes before the $a$ labeling edge $(\pi(v), v)$ also in sequence $OUT$. In our example, those are the two 'a' contained in $OUT[1]$ and $OUT[6]$. We finally obtain our claim: equally-labeled edges appear in the same relative order in $IN$ and $OUT$.

The XBWT is a generalization to labeled trees of a well-known string transform—the Burrows-Wheeler transform (BWT) [20]—described for the first time in 1994 (the BWT is precisely sequence $OUT$ of a path tree (To be precise, the original BWT used the symmetric lexicographic order of the string's suffixes). Its corresponding index, the FM-index [16] was first described by Ferragina and Manzini in 2000. The property we just described—allowing the mapping of characters from $IN$ to $OUT$—is the building block of the FM-index and takes the name *LF mapping*. The LF mapping can be used to perform a visit of the tree using just $OUT$. First, note that $OUT$ fully specifies $IN$: the latter is a sorted list of all characters

appearing in *OUT*, plus character $. Start from the virtual incoming edge of the root, which appears always in $IN[1]$. The outgoing labels of the roots appear in $OUT[1]$. Using the LF property, we map the outgoing labels of the root to sequence *IN*, obtaining the locations in PA of the successors of the root. The reasoning can be iterated, ultimately leading to a complete visit of the tree.

### 3.5.4. Subpath Queries

The LF mapping can also be used to answer counting queries. Suppose we wish to count the number of nodes reached by a path labeled with string $\Pi[1, m]$. For example, consider the tree of Figure 7 and let $\Pi = aa$. First, find the range $PA[\ell, r]$ of nodes on PA reached by $\Pi[1]$. This is easy, since characters in *IN* are sorted: $[\ell, r]$ is the maximal range such that $IN[i] = \Pi[1]$ for all $\ell \leq i \leq r$. In our example, $\ell = 2$ and $r = 5$. Now, note that $OUT[\ell, r]$ contains the outgoing labels of all nodes reached by $\Pi[1]$. Then, we can extend our search by one character by following all edges in $OUT[\ell, r]$ labeled with $\Pi[2]$. In our example, $\Pi[2] = a$, and there are 2 edges labeled with 'a' to follow: those at positions $OUT[2]$ and $OUT[3]$. This requires applying the LF mapping to all those edges. Crucially, note that the LF mapping property also guarantees that the nodes we reach by following those edges form a contiguous co-lexicographic range $PA[\ell', r']$. In our example, we obtain the range $PA[3, 4]$, containing pre-order nodes 3 and 4. These are precisely the $occ = \ell' - r + 1$ nodes reached by $\Pi[1, 2]$ (and $occ$ is the answer to the count query for $\Pi[1, 2]$). It is clear that the reasoning can be iterated until finding the range of all nodes reached by a pattern $\Pi$ of any length.

For clarity, in our discussion above, we have ignored efficiency details. If implemented as described, each extension step would require $O(n)$ time (a linear scan of *IN* and *OUT*). It turns out that, using up-to-date data structures [69], a single character-extension step can be implemented in just $O\left(\log\left(\frac{\log \sigma}{\log n}\right)\right)$ time! The idea is, given the range $PA[\ell, r]$ of $\Pi[1]$, to locate just the first and last occurrence of $\Pi[2]$ in $OUT[\ell, r]$. This can be implemented with a so-called *rank* query (that is, the number of characters equal to $\Pi[2]$ before a given position $OUT[i]$). We redirect the curious reader to the original articles by Ferragina et al. [35,68] (XBWT) and Belazzougui and Navarro [69] (up-to-date rank data structures) for the exquisite data structure details.

As far as locate queries are concerned, as previously mentioned a simple strategy could be to explicitly store PA. In the above example, the result of *locate* for pattern $\Pi = aa$ would be the pre-order nodes $PA[3, 4] = 3$, 4. This solution, however, uses $n \log n$ bits on top of the XBWT. Arroyuelo et al. [70] Section 5.1 and Prezza [49] describe a sampling strategy that solves this problem when the nodes' identifiers are DFS numbers: fix a parameter $t \leq n$. The idea is to decompose the tree into $\Theta(n/t)$ subtrees of size $O(t)$, and explicitly store in $O((n/t) \log n)$ bits the pre-order identifiers of the subtrees' roots. Then, the pre-order of any non-sampled node can be retrieved by performing a visit of a subtree using the XBWT navigation primitives. By fixing $t = \log^{1+\epsilon} n$ for any constant $\epsilon > 0$, this strategy can compute any $PA[i]$ (i.e., locate any pattern occurrence) in polylogarithmic time while using just $o(n)$ bits of additional space. A more advanced mechanism [49] allows locating the DFS number of each pattern occurrence in optimal $O(1)$ time within compressed space.

### 3.5.5. Compression

To conclude, we show how the XBWT (and, similarly, the BWT of a string) can be compressed. In order to efficiently support the LF mapping, sequence *IN* has to be explicitly stored. However, note that this sequence is strictly increasing. If the alphabet is effective and of the form $\Sigma = [1, \sigma]$, then we can represent *IN* with a simple bitvector of $n$ bits marking with a bit '1' the first occurrence of each new character. If $\sigma$ is small (for example, polylog($n$)), then this bitvector can be compressed down to $o(n)$ bits while supporting efficient queries [71]. Finally, we concatenate all characters of *OUT* in a single string and use another bitvector of $2n$ bits to mark the borders between each $OUT[i]$ in

the sequence. Still assuming a small alphabet and using up-to-date data structures [69], this representation takes $nH_0 + 2n + o(n)$ bits of space and supports optimal-time count queries. In their original article, Ferragina et al. [35] realized that this space could be further improved: since nodes are sorted in co-lexicographic order, they are also clustered by the paths connecting them to the root. Then, for a sufficiently small $k \in O(\log_\sigma n)$, we can partition *OUT* by all distinct paths of length $k$ that reach the nodes, and use a different zero-order compressor for each class of the partition. This solution achieves high-order compressed space $nH_k + 2n + o(n)$.

Another method to compress the XBWT is to exploit repetitions of isomorphic subtrees. Alanko et al. [48] show how this can be achieved by a technique they call *tunneling* and that consists in collapsing isomorphic subtrees that are adjacent in co-lexicographic order. Tunneling works for a more general class of graphs (Wheeler graphs), so we discuss it more in detail in Section 3.7. Similarly, one can observe that a repeated topology will generate long runs of equal sets in *OUT* [49]. Let $r$ be the number of such runs. Repetitive trees (including repetitive texts) satisfy $r \ll n$. In the tree in Figure 7, we have $n = 9$ and $r = 7$. For an example of a more repetitive tree, see Reference [49].

*3.6. Further Generalizations*

As we have seen, prefix sorting generalizes quite naturally to labeled trees. There is no reason to stop here: trees are just a particular graph topology. A few years after the successful XBWT was introduced, Mantaci et al. [72–74] showed that finite sets of circular strings, i.e., finite collections of disjoint labeled cycles, do enjoy the same *prefix-sortability* property: the nodes of these particular labeled graphs can be arranged by the strings labeling their incoming paths, thus speeding up subsequent substring searches on the graph. Seven years later, Bowe et al. [75] added one more topology to the family: de Bruijn graphs. A de Bruijn graph of order $k$ for a string $S$ (or for a collection of strings; the generalization is straightforward) has one node for each distinct substring of length $k$ (a $k$-mer) appearing in $S$. Two nodes, representing $k$-mers $s_1$ and $s_2$, are connected if and only if they overlap by $k - 1$ characters (that is, the suffix of length $k - 1$ of $s_1$ equals the prefix of the same length of $s_2$) *and* their concatenation of length $k + 1$ is a substring of $S$. The topology of a de Bruijn graph could be quite complex; in fact, one could define such a graph starting from the $k$-mers read on the paths of an arbitrary labeled graph. For large enough $k$, the set of strings read on the paths of the original graph and the derived de Bruijn graph coincide. Sirén et al. [76] exploited this observation to design a tool to index pan-genomes (that is, genome collections represented as labeled graphs). Given an arbitrary input labeled graph, their GCSA (Generalized Compressed Suffix Array) builds a de Bruijn graph that is equivalent to (i.e., in which paths spell the same strings of) the input graph. While this idea allows to index arbitrary labeled graphs, in the worst case, this conversion could generate an exponentially-larger de Bruijn graph (even though they observe that, in practice, due to the particular distribution of DNA mutations, this exponential explosion does not occur too frequently in bioinformatics applications). A second, more space-efficient version of their tool [77] fixes the maximum order $k$ of the target de Bruijn graph, thus indexing only paths of length at most $k$ of the input graph. Finally, *repeat-free founder block graphs* [78] are yet another recent class of indexable labeled graphs.

In this survey, we do not enter into the details of the above generalizations since they are all particular cases of a larger class of labeled graphs which will be covered in the next subsection: Wheeler graphs. As a result, the search mechanism for Wheeler graphs automatically applies to those particular graph topologies. We redirect the curious reader to the seminal work of Gagie et al. [50] for more details of how these graphs, as well as several other combinatorial objects (FM index of an alignment [79,80], Positional BWT [81], wavelet matrices [82], and wavelet trees [83]), can be elegantly described in the Wheeler graphs framework. In turn, in Section 3.8, we will see that Wheeler graphs are just the "base case" of a larger family of prefix-sortable graphs, ultimately encompassing all labeled graphs: $p$-sortable graphs.

### 3.7. Wheeler Graphs

In 2017, Gagie et al. [50] generalized the principle underlying the approaches described in the previous sections to all labeled graphs in which nodes can be sorted co-lexicographically in a total order. They called such objects *Wheeler graphs* in honor of David J. Wheeler, one of the two inventors of the ubiquitous Burrows-Wheeler transform [20] that today stands at the heart of the most successful compressors and compressed text indexes.

Recall that, for convenience, we treat the special (not too restrictive) case of labeled graphs corresponding to the state transition of finite automata. Indexing finite automata allows us to generalize the ideas behind suffix arrays to (possibly infinite) sets of strings: all the strings read from the source to any of the automaton's final states. Said otherwise, an index for a finite automaton is capable of recognizing the substring closure of the underlying regular language. Consider the following regular language:

$$\mathcal{L} = (\epsilon|aa)b(ab|b)^*,$$

as well as consider the automaton recognizing $\mathcal{L}$ depicted in Figure 9.

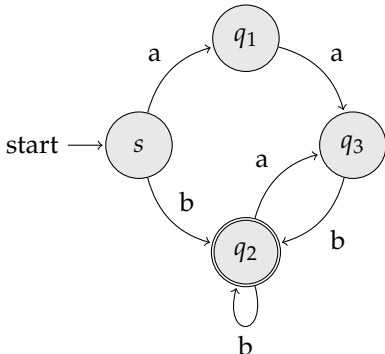

**Figure 9.** Automaton recognizing the regular language $\mathcal{L} = (\epsilon|aa)b(ab|b)^*$.

Let $u$ be a state, and denote with $I_u$ the (possibly infinite) set of all strings read from the source to $u$. Following the example reported in Figure 9, we have $I_{q_1} = \{a\}$, $I_{q_2} = \{b, bb, bab, babb, aab, \dots\}$, and $I_{q_3} = \{aa, aaba, aabba, \dots\}$. Intuitively, if the automaton is a DFA, then we are going to sort its states in such a way that two states are placed in the order $u < v$ if and only if all the strings in $I_u$ are co-lexicographically smaller than all the strings in $I_v$ (for NFAs, the condition is slightly more involved as those sets can have a nonempty intersection). Note that, at this point of the discussion, it is not yet clear that such an ordering *always* exists (in fact, we will see that it does not); however, from the previous subsections, we know that particular graph topologies (in particular, paths, cycles, trees, and de Bruijn graphs) do admit a solution to this sorting problem. The legitimate question, tackled for the first time in Gagie et al.'s work, is: what is the largest graph family admitting such an ordering?

Let $a, a'$ be two characters labeling edges $(u, u')$ and $(v, v')$, respectively. We require the following three *Wheeler properties* [50] for our ordering $\leq$:

(i)     all states with in-degree zero come first in the ordering,
(ii)    if $a < a'$, then $u' < v'$, and
(iii)   if $a = a'$ and $u < v$, then $u' \leq v'$.

It is not hard to see that (i)–(iii) generalize the familiar co-lexicographic order among the prefixes of a string to labeled graphs. In this generalized context, we deal with prefixes of the recognized language, i.e., with the strings labeling paths connecting the source state with any other state. Note also that rule (ii) implicitly requires that all labels entering a state must be equal. This is not a severe restriction: at the cost of increasing the number of states by a factor $\sigma$, any automaton can be transformed into an equivalent one with this property [65]. Figure 10 depicts the automaton of Figure 9 after having ordered (left-to-

right) the states according to the above three Wheeler properties.

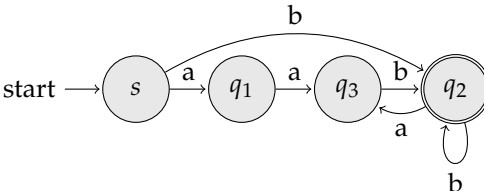

**Figure 10.** The automaton of Figure 9, sorted according to the three Wheeler properties.

An ordering $\leq$ satisfying Wheeler properties (i)–(iii) is called a *Wheeler order*. Note that a Wheeler order, when it exists, is always total (this will become important in Section 3.8). An automaton (or a labeled graph) is said to be Wheeler if its states admit at least one Wheeler order.

### 3.7.1. Subpath Queries

When a Wheeler order exists, all the power of suffix arrays can be lifted to labeled graphs: (1) the nodes reached by a path labeled with a given pattern $\Pi$ form a consecutive range in Wheeler order, and (2) such a range can be found in linear time as a function of the pattern's length. In fact, the search algorithm generalizes those devised for the particular graphs discussed in the previous sections. Figure 11 depicts the process of searching all nodes reached by a path labeled with pattern $\Pi = $ "aba". The algorithm starts with $\Pi = \epsilon$ (empty string, all nodes) and right-extends it by one letter step by step, following the edges labeled with the corresponding character of $\Pi$ from the current set of nodes. Note the following crucial property: at each step, the nodes reached by the current pattern form a *consecutive range*. This makes it possible to represent the range in constant space (by specifying just the first and last node in the range). Similarly to what we briefly discussed in Section 3.5 (*Subpath Queries on the XBWT*), by using appropriate compressed data structures supporting rank and select on strings, each character extension can be implemented efficiently (i.e., in log-logarithmic time). The resulting data structure can be stored in entropy-compressed space [50].

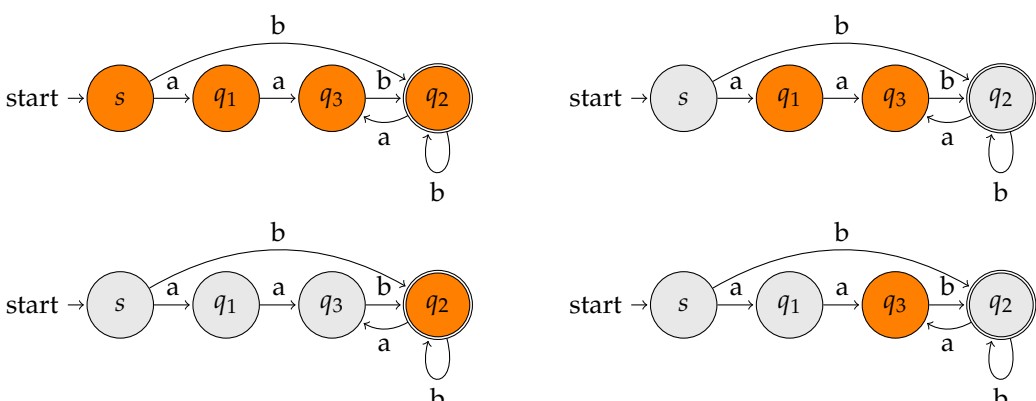

**Figure 11.** Searching nodes reached by a path labeled "aba" in a Wheeler graph. Top left: we begin with the nodes reached by the empty string (full range). Top right: range obtained from the previous one following edges labeled 'a'. Bottom left: range obtained from the previous one following edges labeled 'b'. Bottom right: range obtained from the previous one following edges labeled 'a'. This last range contains all nodes reached by a path labeled "aba"

The algorithm we just described allows us to find the number of nodes reached by a given pattern, i.e., to solve *count* queries on the graph. As far as *locate* queries are concerned, Sirén et al. in Reference [76] proposed a sampling scheme that returns the labels of all nodes reached by the query pattern, provided that the labeling scheme satisfies a

particular monotonicity property (that is, the labels of a node are larger than those of its predecessors). In practical situations, such as genome-graph indexing, such labels can be equal to an absolute position on a reference genome representing all the paths of the graph. In the worst case, however, this scheme requires to store $\Theta(n)$ labels of $O(\log n)$ bits each. As another option, the sampling strategy described by Arroyuelo et al. [70] Section 5.1 and Prezza [49] (see Section 3.5.4 for more details) can be directly applied to the DFS spanning forest of any Wheeler graph, thus yielding locate queries in polylogarithmic time and small additiona redundancy.

### 3.7.2. Compression

The nodes of a Wheeler graph are sorted (clustered) with respect to their incoming paths, so the outgoing labels of adjacent nodes are likely to be similar. This allows applying high-order compression to the labels of a Wheeler graph [50]. It is even possible to compress the graph's topology (together with the labels), if this is highly repetitive (i.e., the graph has large repeated isomorphic subgraphs). By generalizing the *tunneling* technique originally devised by Baier [84] for the Burrows-Wheeler transform, Alanko et al. [48] showed that isomorphic subtrees adjacent in co-lexicographic order can be collapsed while maintaining the Wheeler properties. In the same paper, they showed that a *tunneled* Wheeler graph can be even indexed to support existence path queries (a relaxation of counting queries where we can only discover whether or not a query pattern labels some path in the graph). Similarly, one can observe that a repetitive graph topology will generate long runs of equal sets of outgoing labels in Wheeler order. This allows applying run-length compression to the Wheeler graph [49]. Run-length compression of a Wheeler graph is equivalent to collapsing isomorphic subgraphs that are adjacent in Wheeler order and, thus, shares many similarities with the tunneling technique. A weaker form of run-length compression of a Wheeler graph has been considered by Bentley et al. in Reference [85]. In this work, they turn a Wheeler graph into a string by concatenating the outgoing labels of the sorted nodes, permuted so as to minimize the number of equal-letter runs of the resulting string. They show that the (decision version of the) problem of finding the ordering of the Wheeler graph's sources that minimizes the number of runs is NP-complete. In the same work, they show that also the problem of finding the alphabet ordering minimizing the number of equal-letter runs in the BWT is NP-complete.

### 3.7.3. Sorting and Recognizing Wheeler Graphs

Perhaps not surprisingly, not all labeled graphs admit a Wheeler order of their nodes. Intuitively, this happens because of conflicting predecessors: if $u$ and $v$ have a pair of predecessors ordered as $u' < v'$ and another pair ordered as $v'' < u''$, then a Wheeler order cannot exist. As an example, consider any automaton recognizing the language $\mathcal{L}' = (ax^*b)|(cx^*d)$ (original example by Gagie et al. [50]). Since any string beginning with letter 'a' must necessarily end with letter 'b' and, similarly, any string beginning with letter 'c' must necessarily end with letter 'd', the paths spelling $ax^kb$ and $ax^{k'}d$ must be disjoint for all $k, k' \geq 0$. Said otherwise, the nodes reached by label 'x' and leading to 'b' must be disjoint from those reached by label 'x' and leading to 'd'. Denote with $u_\alpha$ a state reached by string $\alpha$ (read from the source). Then, it must be $u_{ax} < u_{bx}$: from the above observation, those two states must be distinct, and, from the Wheeler properties, they must be in this precise order. For the same reason, it must be the case that $u_{bx} < u_{axx} < u_{bxx} < u_{axxx} < \ldots$. This is an infinite sequence of distinct states: no finite Wheeler automaton can recognize $\mathcal{L}'$.

This motivates the following natural questions: given an automaton $\mathcal{A}$, is it Wheeler? if yes, can we efficiently find a corresponding Wheeler order? It turns out that these are hard problems. Gibney and Thankachan showed in Reference [86] that deciding whether an arbitrary automaton admits a Wheeler order is NP-complete. This holds even when each state is allowed to have at most five outgoing edges labeled with the same character (which somehow bounds the amount of nondeterminism). On the positive side, Alanko et al. [65] showed that both problems (recognition and sorting) can be solved in quadratic time when

each state is allowed to have at most two outgoing edges labeled with the same character. This includes DFAs, for which, however, a more efficient linear-time solution exists [65]. The complexity of the cases in between, i.e., at most three/four equally-labeled outgoing edges, is still open [87].

### 3.7.4. Wheeler Languages

Since we are talking about finite automata, one question should arise naturally: what languages are recognized by Wheeler NFAs? Let us call *Wheeler languages* this class. First of all, Wheeler languages are clearly regular since, by definition, they are accepted by finite state automata. Moreover, all finite languages are Wheeler because they can be recognized by a tree-shaped automaton, which (as seen in Section 3.5) is always prefix-sortable. Additionally, as observed by Gagie et al. [50] not all regular languages are Wheeler: $(ax^*b)|(cx^*d)$ is an example. Interestingly, Wheeler languages can be defined both in terms of DFAs and NFAs: Alanko et al. proved in Reference [65] that Wheeler NFAs and Wheeler DFAs recognize the same class of languages. Another powerful language-theoretic result that can be transferred from regular to Wheeler languages is their neat algebraic characterization based on Myhill-Nerode equivalence classes [66]. We recall that two strings $\alpha$ and $\beta$ are Myhill-Nerode equivalent with respect to a given regular language $\mathcal{L}$ if and only if, for any string $\gamma$, we have that $\alpha\gamma \in \mathcal{L} \Leftrightarrow \beta\gamma \in \mathcal{L}$. The Myhill-Nerode theorem states that $\mathcal{L}$ is regular if and only if the Myhill-Nerode equivalence relation has finite index (i.e., it has a finite number of equivalence classes). In the Wheeler case, the Myhill-Nerode equivalence relation is slightly modified by requiring that equivalence classes of prefixes of the language are also intervals in co-lexicographic order and contain words ending with the same letter. After this modification, the Myhill-Nerode theorem can be transferred to Wheeler languages: $\mathcal{L}$ is Wheeler if and only if the modified Myhill-Nerode equivalence relation has finite index [65,88]. See Alanko et al. [88] for a comprehensive study of the elegant properties of Wheeler languages (including closure properties and complexity of the recognition problem). From the algorithmic point of view (which is the most relevant to this survey), these results permit to define and build efficiently the *minimum* Wheeler DFA (that is, with the smallest number of states) recognizing the same language of a given input Wheeler DFA, thus optimizing the space of the resulting index [65].

After this introduction to Wheeler languages, let us consider the question: can we index Wheeler languages efficiently? Interestingly, Gibney and Thankachan's NP-completeness proof [86] requires that the automaton's topology is fixed so it does not rule out the possibility that we can index in polynomial time an equivalent automaton. After all, in many situations, we are actually interested in indexing a language rather than a fixed graph topology. Surprisingly, the answer to the above question is yes: it is easier to index Wheeler languages, rather than Wheeler automata. Since recognizing and sorting Wheeler DFAs is an easy problem, a first idea could be to turn the input NFA into an equivalent DFA. While it is well-known that, in the worst case, this conversion (via the powerset algorithm) could result in an exponential blow-up of the number of states, Alanko et al. [65] proved that a Wheeler NFA always admits an equivalent Wheeler DFA of linear size (the number of states doubles at most). This has an interesting consequence: if the input NFA is Wheeler, then we can index its language in linear time (in the size of the input NFA) after converting it to a DFA (which requires polynomial time). We can actually do better: if the input NFA $\mathcal{A}$ is Wheeler, then in polynomial time we can build a Wheeler NFA $\mathcal{A}'$ that (i) is never larger than $\mathcal{A}$, (ii) recognizes the same language as $\mathcal{A}$, and (iii) can be sorted in polynomial time [88]. We remark that there is a subtle reason why the above two procedures for indexing Wheeler NFAs do not break the NP-completeness of the problem of recognizing this class of graphs: it is possible that they generate a Wheeler NFA even if the input is not a Wheeler NFA; thus they cannot be used to solve the recognition problem. To conclude, these strategies can be used to index Wheeler NFAs, but do not tell us anything about indexing Wheeler languages represented as a general (possibly non-Wheeler) NFA. An interesting case is represented by finite languages (always Wheeler)

represented as acyclic NFAs. The lower bounds discussed in Section 3.2 tell us that a quadratic running time is unavoidable for the graph indexing problem, even in the acyclic case. This implies, in particular, that the conversion from arbitrary acyclic NFAs to Wheeler NFAs must incur in a quadratic blow-up in the worst case. In practice, the situation is much worse: Alanko et al. showed in Reference [65] that the blow-up is exponential in the worst case. In the same paper, they provided a fast algorithm to convert any acyclic DFA into the smallest equivalent Wheeler DFA.

### 3.8. p-Sortable Automata

Despite its power, the Wheeler graph framework does not allow to index and compress *any* labeled graph: it is easy to come up with arbitrarily large Wheeler graphs that lose their Wheeler property after the addition of just *one* edge. Does this mean that such an augmented graph cannot be indexed efficiently? clearly not, since it would be sufficient to add a small "patch" (keeping track of the extra edge) to the index of the underlying Wheeler subgraph in order to index it. As another "unsatisfying" example, consider the union $\mathcal{L} = \mathcal{L}_1 \cup \mathcal{L}_2$ of two Wheeler languages $\mathcal{L}_1$ and $\mathcal{L}_2$. In general, $\mathcal{L}$ is not a Wheeler language [88]. However, $\mathcal{L}$ can be easily indexed by just keeping two indexes (of two Wheeler automata recognizing $\mathcal{L}_1$ and $\mathcal{L}_2$), *with no asymptotic slowdown* in query times! The latter example is on the right path to a solution of the problem. Take two Wheeler automata $\mathcal{A}_1$ and $\mathcal{A}_2$ recognizing two Wheeler languages $\mathcal{L}_1$ and $\mathcal{L}_2$, respectively, such that $\mathcal{L} = \mathcal{L}_1 \cup \mathcal{L}_2$ is not Wheeler. The union automaton $\mathcal{A}_1 \cup \mathcal{A}_2$ (obtained by simply merging the start states of $\mathcal{A}_1$ and $\mathcal{A}_2$) is a nondeterministic automaton recognizing $\mathcal{L}$. Taken individually, the states of $\mathcal{A}_1$ and $\mathcal{A}_2$ can be sorted in two total orders. However, taken as a whole, the two sets of states do not admit a total co-lexicographic order (which would imply that $\mathcal{L}$ is Wheeler). The solution to this riddle is to abandon total orders in favor of *partial orders* [89]. A partial order $\leq$ on a set $V$ (in our case, the set of the automaton's states) is a reflexive, antisymmetric and transitive relation on $V$. In a partial order, two elements $u, v$ either are comparable, in which case $u \leq v$ or $v \leq u$ hold (both hold only if $u = v$), or are not comparable, in which case neither $u \leq v$ nor $v \leq u$ hold. The latter case is indicated as $u \parallel v$. Our co-lexicographic partial order is defined as follows. Let $a, a'$ be two characters labeling edges $(u, u')$ and $(v, v')$, respectively. We require the following properties:

(i)   all states with in-degree zero come first in the ordering,
(ii)  if $a < a'$, then $u' < v'$, and
(iii) if $a = a'$ and $u' < v'$, then $u \leq v$.

Note that, differently from the definition of Wheeler order (Section 3.7), the implication of (iii) follows the edges *backwards*. As it turns out, $\leq$ is indeed a partial order and, as we show in the next subsections, it allows generalizing the useful properties of Wheeler graphs to arbitrary topologies. A convenient representation for any partial order is a Hasse diagram: a directed acyclic graph where we draw the elements of the order from the smallest (bottom) to largest ones (top), and two elements are connected by an edge $(u, v)$ if and only if $u \leq v$. See Figure 12 for an example.

The lower bounds discussed in Section 3.2 tell us that indexed pattern matching on graphs cannot be solved faster than quadratic time in the worst case. In particular, this means that our generalization of Wheeler graphs cannot yield an index answering subpath queries in linear time. In fact, there is a catch: the extent to which indexing and compression can be performed efficiently is proportional to the similarity of the partial order $\leq$ to a total one. As it turns out, the correct measure of similarity to consider is the *order's width*: the minimum number $p$ of totally-ordered chains (subsets of states) into which the set of states can be partitioned. Figure 12 makes it clear that the states of the automaton can be divided into $p = 2$ totally-ordered subsets. This choice is not unique (look at the Hasse diagram), and a possible partition is $s < q_1 < q_3$ (in yellow) and $q_2$ (in red). We call *p*-sortable the class of automata for which there exists a chain partition of size $p$. Since the $n$ states of any automaton can always be partitioned into $n$ chains, this definition captures

all automata. The parameter $p$ seems to capture some deep regularity of finite automata: in addition to determining the compressibility of their topology (read next subsections), it also determines their inherent determinism: a $p$-sortable NFA with $n$ states always admits an equivalent DFA (which can be obtained via the standard powerset algorithm) with at most $2^p(n - p + 1) - 1$ states. This represents some sort of fine-grained analysis refining the folklore (tight) bound of $2^n$ [90] and has deep implications to several algorithms on automata. For example, it implies that the PSPACE-complete NFA equivalence problem is fixed-parameter tractable with respect to $p$ [89]. To conclude, we mention that finding the smallest $p$ for a given labeled graph is NP complete, though the problem admits an $O(e^2 + n^{5/2})$-time algorithm for the deterministic case [89].

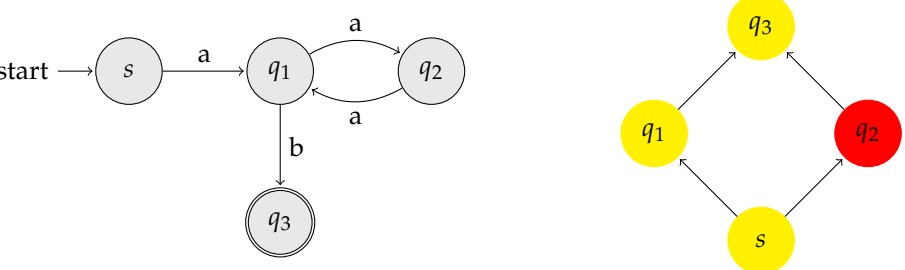

**Figure 12.** (**Left**) Automaton for the language $\mathcal{L} = a(aa)^*b$. (**Right**) Hasse diagram of a co-lexicographic partial order for the graph. The order's width is 2, and the order can be partitioned into two totally-ordered chains (yellow and red in the example; this is not the only possible choice).

### 3.8.1. Subpath Queries

Subpath queries can be answered on $p$-sortable automata by generalizing the forward search algorithm of Wheeler graphs. In fact, the following property holds: for any pattern $\Pi$, the states reached by a path labeled with $\Pi$ always form one convex set in the partial order. In turn, any convex set can be expressed as $p$ intervals, one contained in each class of the chain partition [89] (for any chain partition). The generalized forward search algorithm works as follows: start with the interval of $\epsilon$ on the $p$ chains (the full interval on each chain). For each character $a$ of the pattern, follow the edges labeled with $a$ that depart from the current (at most) $p$ intervals. By the above property, the target nodes will still be contained in (at most) $p$ intervals on the chains. Consider the example of Figure 12. In order to find all nodes reached by pattern 'aa', the generalized forward search algorithm first finds the nodes reached by 'a': those are nodes $q_1$ (a range on the yellow chain) and $q_2$ (a range on the red chain). Finally, the algorithm follows all edges labeled 'a' from those ranges, thereby finding all nodes reached by 'aa': nodes (again) $q_1$ and $q_2$ which, as seen above, form one range on the yellow chain and one on the red chain. See [89] for a slightly more involved example. Using up-to-date data structures, the search algorithm can be implemented to run in $O(|\Pi| \cdot \log(\sigma p) \cdot p^2)$ steps [89]. In the worst case ($p = n$), this essentially matches the lower bound of Equi et al. [52–54] on dense graphs (see Section 3.2). On the other hand, for small values of $p$ this running time can be significantly smaller than the lower bound: in the extreme case, $p = 1$, and we obtain the familiar class of Wheeler graphs (supporting optimal-time subpath queries).

As far as locate queries are concerned, the sampling strategy described by Arroyuelo et al. [70] Section 5.1 and Prezza [49] (see Section 3.5.4 for more details) can be directly applied to the DFS spanning forest of any $p$-sortable graph (similarly to Wheeler graphs), thus yielding locate queries in polylogarithmic time and small additional redundancy. This is possible since the index for $p$-sortable graphs described in Reference [89] supports navigation primitives, as well, and can thus be used to navigate the DFS spanning forest of the graph.

### 3.8.2. Compression

The Burrows-Wheeler transform [20] (also see Section 3.5) can be generalized to $p$-

sortable automata: the main idea is to (i) partition the states into $p$ totally-sorted chains (for example, in Figure 12, one possible such chain decomposition is $\{s, q_1, q_3\}, \{q_2\}$), (ii) order the states by "gluing" the chains in any order (for example, in Figure 12 one possible such ordering is $s, q_1, q_3, q_2$), (iii) build the adjacency matrix of the graph using this state ordering and (iv) for each edge in the matrix, store only its label and its endpoint chains (that is, two numbers between 1 and $p$ indicating which chains the two edges' endpoints belong to). It can be shown that this matrix can be linearized in an invertible representation taking $\log \sigma + 2 \log p + 2 + o(1)$ bits per edge. On DFAs, the representation takes less space: $\log \sigma + \log p + 2 + o(1)$ bits per edge [89]. This is already a compressed representation of the graph, since "sortable" graphs (i.e., having small $p$) are compressed to few bits per edge, well below the information-theoretic worst-case lower bound if $p \ll n$. Furthermore, within each chain the states are sorted by their incoming paths. It follows that, as seen in Sections 3.5 and 3.7, high-order entropy-compression can be applied within each chain.

## 4. Conclusions and Future Challenges

In this survey, we tried to convey the core ideas that have been developed to date in the field of compressed graph indexing, in the hope of introducing (and attracting) the non-expert reader to this exciting and active area of research. As it can be understood by reading our survey, the field is in rapid expansion and does not lack of stimulating challenges. On the lower-bound side, a possible improvement could be to provide fine-grained lower bounds to the graph indexing problem, e.g., as a function of the parameter $p$ introduced in Section 3.8. As far as graph compression is concerned, it is plausible that new powerful graph compression schemes will emerge from developments in the field: an extension of the run-length Burrows Wheeler transform to labeled graphs, for example, could compete with existing grammar-based graph compressors while also supporting efficient path queries. Efficient index construction is also a considerable challenge. As seen in Section 3.7, Wheeler graphs can be sorted efficiently only in the deterministic case. Even when the nondeterminism degree is very limited to just two equally-labeled edges leaving a node, the fastest sorting algorithm has quadratic complexity. Even worse, in the general (nondeterministic) case, the problem is NP-complete. The situation does not improve with the generalization considered in Section 3.8: finding the minimum width $p$ for a deterministic graph takes super-quadratic time with current solutions, and it becomes NP-complete for arbitrary graphs. Clearly, practical algorithms for graph indexing will have to somehow sidestep these issues. One possible solution could be approximation: we note that, even if the optimal $p$ cannot be found efficiently (recall that $p = 1$ for Wheeler graphs), approximating it (for example, up to a polylog($n$) factor) would still guarantee fast queries. Further challenges include generalizing the labeled graph indexing problem to allow aligning regular expressions against graphs. This could be achieved, for example, by indexing both the graph and an NFA recognizing the query (regular expression) pattern.

**Funding:** This research received no external funding.

**Institutional Review Board Statement:** Not applicable.

**Informed Consent Statement:** Not applicable.

**Data Availability Statement:** Data sharing not applicable.

**Acknowledgments:** I would like to thank Alberto Policriti for reading a preliminary version of this manuscript and providing useful suggestions.

**Conflicts of Interest:** The authors declare no conflict of interest.

**Abbreviations**

The following abbreviations are used in this manuscript:

| | |
|---|---|
| BWT | Burrows-Wheeler Transform |
| DFA | Deterministic Finite Automaton |
| GCSA | Generalized Compressed Suffix Array |
| NFA | Nondeterministic Finite Automaton |
| PA | Prefix array |
| SA | Suffix Array |
| XBWT | eXtended Burrows-Wheeler Transform |
| $\mathcal{T}$ | Text (string to be indexed) |
| $\Pi$ | Pattern to be aligned on the labeled graph |
| $\Sigma$ | Alphabet |
| $\sigma$ | Size of the alphabet: $\sigma = |\Sigma|$ |
| $e$ | Number of edges in a labeled graph |
| $m$ | Pattern length: $m = |\Pi|$ |
| $n$ | Length of a text or number of nodes in a labeled graph |
| $p$ | Number of chains in a chain decomposition of the automaton's partial order (Section 3.8) |

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
