# Peer review of "Subpath Queries on Compressed Graphs: A Survey"

_algorithms, doi:10.3390/a14010014_

Round 1

Reviewer 1 Report

Text indexing started at the beginning og the 1970's and has thus a long history by now.
The lastest advances on compressed indexing for texts have been very recently transfered to graph indexing and the field
is evolving very fast. despite being very new, graph indexing has attracted a lot of interest
and the litterature is becoming quite dense.
The author tries to bring some order on this topic by presenting a quite complete review on the area
of compressed graph indexing and querying.
I recommend to accept this submission which is well written and includes the latest references.

Typos & remarks:
Page 1, lines 19-21: The original Boyer-Moore string matching algorithm does not find all the occurrences
of the pattern in the text in linear time
Page 4, line 120: "Being the DNA a string" please reformulate
Page 6, line 226: finte --> finite
Page 6, line 229: ,\cdots --> ,\cdots,
Page 6, line 230: \cdot\cdots\cdot --> \cdots
Page 6, line 232: ,\cdots --> ,\cdots,
Page 9, line 306: The more recent work --> The more recent works
Page 16, line 590: "the number of BWT runs" please be more precise
Page 18, line 669: \cal L_1 --> \cal L_2 (twice)
Page 18, line 671: \cal L_1 --> \cal L_2
Page 18, line 673: \cal L_1 --> \cal L_2 (twice)
Page 21, line 811: (WSP'96 --> (WSP'96)
Page 21, line 822: report #?
Page 22, line 825: bioinformatics --> Bioinformatics

Author Response

Thank you very much for your comments. Here are my responses:

Page 1, lines 19-21: The original Boyer-Moore string matching algorithm does not find all the occurrences of the pattern in the text in linear time

> Thanks for pointing this out: I was thinking of the Boyer-Moore-Galil's modified version when I wrote that. I corrected the citation.

Page 4, line 120: "Being the DNA a string" please reformulate

> rephrased. 

Page 6, line 226: finte --> finite

> fixed

Page 6, line 229: ,\cdots --> ,\cdots,

> fixed

Page 6, line 230: \cdot\cdots\cdot --> \cdots

> fixed

Page 6, line 232: ,\cdots --> ,\cdots,

> fixed

Page 9, line 306: The more recent work --> The more recent works

> fixed

Page 16, line 590: "the number of BWT runs" please be more precise

> changed to "minimizing the number of equal-letter runs in the BWT"

Page 18, line 669: \cal L_1 --> \cal L_2 (twice)
Page 18, line 671: \cal L_1 --> \cal L_2
Page 18, line 673: \cal L_1 --> \cal L_2 (twice)

> Thanks for spotting these copy-paste typos. Fixed.

Page 21, line 811: (WSP'96 --> (WSP'96)

> fixed.

Page 21, line 822: report #?

> added.

Page 22, line 825: bioinformatics --> Bioinformatics

> fixed.

Reviewer 2 Report

This review introduces the most relevant results about (compressed/compressible) indexes for labeled graphs (I.e., for locating substrings of the strings represented by the graphs). The review is fairly comprehensive, yet it does not overlap with similar reviews focused on indexing linear texts. I appreciate the prose – concise and clear – and the illustrative examples. The review is quite self-contained, since it first presents the results known for linear texts that are useful to understand the idea in the general graph case. The review has, mainly, an algorithmic perspective which is completely coherent with the scope of the journal. There are several self-citations but they are not inappropriate since those results are relevant for the presentation (and, furthermore, they testify that the author is an expert of the field). Extremely informative are the connections (and the discussion) performed by the author among the works here presented. The presentation of the open challenges is interesting, too.

Minor issues:

L22 “in this case” -> “In this case”

L164 function φ -> function ψ

L171 as far as I know the cited aligners are not based on CSA

L226 “finte” -> “finite”

L579 “can even by indexed” -> “can be even indexed”

L669 and following: “\mathcal{L}_1” is repeated twice

Author Response

Thank you very much for your comments. Here are my responses:

L22 “in this case” -> “In this case”

> fixed

L164 function φ -> function ψ

> fixed

L171 as far as I know the cited aligners are not based on CSA

> Here I meant the broader meaning of CSA, which includes any compressed representation of suffix arrays (and thus the FM-index). I agree however this could be misleading, since in the previous section I described CSAs and FM-indexes as separate things. I changed to "compressed indexes".

L226 “finte” -> “finite”

> fixed

L579 “can even by indexed” -> “can be even indexed”

> fixed

L669 and following: “\mathcal{L}_1” is repeated twice

> thanks for noticing: this is a copy paste typo and was repeated in multiple places. It is now fixed.